# Attend, Infer, Repeat:
# Fast Scene Understanding with Generative Models

**S. M. Ali Eslami, Nicolas Heess, Theophane Weber, Yuval Tassa,**
**David Szepesvari, Koray Kavukcuoglu, Geoffrey E. Hinton**
{aeslami,heess,theophane,tassa,dsz,korayk,geoffhinton}@google.com
Google DeepMind, London, UK

## Abstract

We present a framework for efficient inference in structured image models that explicitly reason about objects. We achieve this by performing probabilistic inference using a recurrent neural network that attends to scene elements and processes them one at a time. Crucially, the model itself learns to choose the appropriate number of inference steps. We use this scheme to learn to perform inference in partially specified 2D models (variable-sized variational auto-encoders) and fully specified 3D models (probabilistic renderers). We show that such models learn to identify multiple objects – counting, locating and classifying the elements of a scene – without any supervision, e.g., decomposing 3D images with various numbers of objects in a single forward pass of a neural network at unprecedented speed. We further show that the networks produce accurate inferences when compared to supervised counterparts, and that their structure leads to improved generalization.

## 1   Introduction

The human percept of a visual scene is highly structured. Scenes naturally decompose into *objects* that are arranged in space, have visual and physical properties, and are in functional relationships with each other. Artificial systems that interpret images in this way are desirable, as accurate detection of objects and inference of their attributes is thought to be fundamental for many problems of interest. Consider a robot whose task is to clear a table after dinner. To plan its actions it will need to determine which objects are present, what classes they belong to and where each one is located on the table.

The notion of using structured models for image understanding has a long history (e.g., 'vision as inverse graphics' [4]), however in practice it has been difficult to define models that are: (a) expressive enough to capture the complexity of natural scenes, and (b) amenable to tractable inference. Meanwhile, advances in deep learning have shown how neural networks can be used to make sophisticated predictions from images using little interpretable structure (e.g., [10]). Here we explore the intersection of structured probabilistic models and deep networks. Prior work on deep generative methods (e.g., VAEs [9]) have been mostly unstructured, therefore despite producing impressive samples and likelihood scores their representations have lacked interpretable meaning. On the other hand, structured generative methods have largely been incompatible with deep learning, and therefore inference has been hard and slow (e.g., via MCMC).

Our proposed framework achieves scene interpretation via learned, amortized inference, and it imposes structure on its representation through appropriate partly- or fully-specified generative models, rather than supervision from labels. It is important to stress that by training generative models, the aim is not primarily to obtain good reconstructions, but to produce good representations, in other words to understand scenes. We show experimentally that by incorporating the right kinds of structures, our models produce representations that are more useful for downstream tasks than those produced by VAEs or state-of-the-art generative models such as DRAW [3].

The proposed framework crucially allows for reasoning about the complexity of a given scene (the dimensionality of its latent space). We demonstrate that via an Occam's razor type effect, this makes it possible to discover the underlying causes of a dataset of images in an unsupervised manner. For instance, the model structure will enforce that a scene is formed by a variable number of entities that appear in different locations, but the process of learning will identify what these scene elements look like and where they appear in any given image. The framework also combines high-dimensional distributed representations with directly interpretable latent variables (e.g., affine pose). This combination makes it easier to avoid the pitfalls of models that are too unconstrained (leading to data-hungry learning) or too rigid (leading to failure via mis-specification).

The main contributions of the paper are as follows. First, in Sec. 2 we formalize a scheme for efficient variational inference in latent spaces of variable dimensionality. The key idea is to treat inference as an *iterative* process, implemented as a recurrent neural network that attends to one object at a time, and learns to use an *appropriate number* of inference steps for each image. We call the proposed framework *Attend-Infer-Repeat* (AIR). End-to-end learning is enabled by recent advances in amortized variational inference, e.g., combining gradient based optimization for continuous latent variables with black-box optimization for discrete ones. Second, in Sec. 3 we show that AIR allows for learning of generative models that decompose multi-object scenes into their underlying causes, e.g., the constituent objects, in an unsupervised manner. We demonstrate these capabilities on MNIST digits (Sec. 3.1), overlapping sprites and Omniglot glyphs (appendices H and G). We show that model structure can provide an important inductive bias that is not easily learned otherwise, leading to improved generalization. Finally, in Sec. 3.2 we demonstrate how our inference framework can be used to perform inference for a 3D rendering engine with unprecedented speed, recovering the counts, identities and 3D poses of complex objects in scenes with significant occlusion in a single forward pass of a neural network, providing a scalable approach to 'vision as inverse graphics'.

## 2  Approach

In this paper we take a Bayesian perspective of scene interpretation, namely that of treating this task as inference in a generative model. Thus given an image $\mathbf{x}$ and a model $p_\theta^x(\mathbf{x}|\mathbf{z})p_\theta^z(\mathbf{z})$ parameterized by $\theta$ we wish to recover the underlying scene description $\mathbf{z}$ by computing the posterior $p(\mathbf{z}|\mathbf{x}) = p_\theta^x(\mathbf{x}|\mathbf{z})p_\theta^z(\mathbf{z})/p(\mathbf{x})$. In this view, the prior $p_\theta^z(\mathbf{z})$ captures our assumptions about the underlying scene, and the likelihood $p_\theta^x(\mathbf{x}|\mathbf{z})$ is our model of how a scene description is rendered to form an image. Both can take various forms depending on the problem at hand and we will describe particular instances in Sec. 3. Together, they define the language that we use to describe a scene.

Many real-world scenes naturally decompose into objects. We therefore make the modeling assumption that the scene description is structured into groups of variables $\mathbf{z}^i$, where each group describes the attributes of one of the objects in the scene, e.g., its type, appearance, and pose. Since the number of objects will vary from scene to scene, we assume models of the following form:

$$p_\theta(\mathbf{x}) = \sum_{n=1}^{N} p_N(n) \int p_\theta^z(\mathbf{z}|n) p_\theta^x(\mathbf{x}|\mathbf{z}) \mathrm{d}\mathbf{z}. \tag{1}$$

This can be interpreted as follows. We first sample the number of objects $n$ from a suitable prior (for instance a Binomial distribution) with maximum value $N$. The latent, variable length, scene descriptor $\mathbf{z} = (\mathbf{z}^1, \mathbf{z}^2, \ldots, \mathbf{z}^n)$ is then sampled from a scene model $\mathbf{z} \sim p_\theta^z(\cdot|n)$. Finally, we render the image according to $\mathbf{x} \sim p_\theta^x(\cdot|\mathbf{z})$. Since the indexing of objects is arbitrary, $p_\theta^z(\cdot)$ is exchangeable and $p_\theta^x(\mathbf{x}|\cdot)$ is permutation invariant, and therefore the posterior over $\mathbf{z}$ is exchangeable.

The prior and likelihood terms can take different forms. We consider two scenarios: For 2D scenes (Sec. 3.1), each object is characterized in terms of a learned distributed continuous representation for its shape, and a continuous 3-dimensional variable for its pose (position and scale). For 3D scenes (Sec. 3.2), objects are defined in terms of a categorical variable that characterizes their identity, e.g., sphere, cube or cylinder, as well as their positions and rotations. We refer to the two kinds of variables for each object $i$ in both scenarios as $\mathbf{z}_{\text{what}}^i$ and $\mathbf{z}_{\text{where}}^i$ respectively, bearing in mind that their meaning (e.g., position and scale in pixel space vs. position and orientation in 3D space) and their data type (continuous vs. discrete) will vary. We further assume that $\mathbf{z}^i$ are independent under the prior, i.e., $p_\theta^z(\mathbf{z}|n) = \prod_{i=1}^{n} p_\theta^z(\mathbf{z}^i)$, but non-independent priors, such as a distribution over hierarchical scene graphs (e.g., [28]), can also be accommodated. Furthermore, while the number of objects is bounded as per Eq. 1, it is relatively straightforward to relax this assumption.

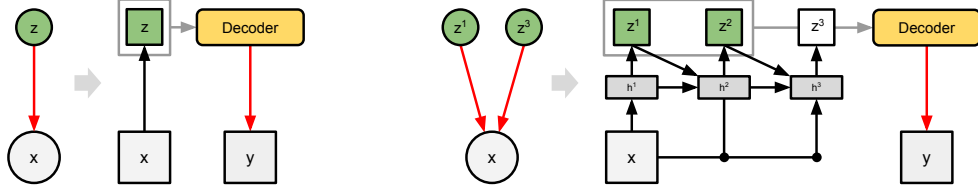

Figure 1: *Left:* A single random variable $z$ produces the observation $x$ (the image). The relationship between $z$ and $x$ is specified by a model. Inference is the task of computing likely values of $z$ given $x$. Using an auto-encoding architecture, the model (red arrow) and its inference network (black arrow) can be trained end-to-end via gradient descent. *Right:* For most images of interest, multiple latent variables (e.g., multiple objects) give rise to the image. We propose an iterative, variable-length inference network (black arrows) that attends to one object at a time, and train it jointly with its model. The result is fast, feed-forward, interpretable scene understanding trained without supervision.

## 2.1 Inference

Despite their natural appeal, inference for most models in the form of Eq. 1 is intractable due to the dimensionality of the integral. We therefore employ an amortized variational approximation to the true posterior by learning a distribution $q_\phi(\mathbf{z}, n|\mathbf{x})$ parameterized by $\phi$ that minimizes $\mathrm{KL}\left[q_\phi(\mathbf{z}, n|\mathbf{x})||p_\theta^z(\mathbf{z}, n|\mathbf{x})\right]$. While such approximations have recently been used successfully in a variety of works [21, 9, 18] the specific form of our model poses two additional difficulties. *Trans-dimensionality:* As a challenging departure from classical latent space models, the size of the latent space $n$ (i.e., the number of objects) is a random variable itself, which necessitates evaluating $p_N(n|\mathbf{x}) = \int p_\theta^z(\mathbf{z}, n|x)\mathrm{d}\mathbf{z}$, for all $n = 1...N$. *Symmetry:* There are strong symmetries that arise, for instance, from alternative assignments of objects appearing in an image $\mathbf{x}$ to latent variables $\mathbf{z}^i$.

We address these challenges by formulating inference as an iterative process implemented as a recurrent neural network, which infers the attributes of one object at a time. The network is run for $N$ steps and in each step explains one object in the scene, conditioned on the image and on its knowledge of previously explained objects (see Fig. 1).

To simplify sequential reasoning about the number of objects, we parameterize $n$ as a variable length latent vector $\mathbf{z}_{\mathrm{pres}}$ using a unary code: for a given value $n$, $\mathbf{z}_{\mathrm{pres}}$ is the vector formed of $n$ ones followed by one zero. Note that the two representations are equivalent. The posterior takes the following form:

$$q_\phi(\mathbf{z}, \mathbf{z}_{\mathrm{pres}}|\mathbf{x}) = q_\phi(z_{\mathrm{pres}}^{n+1} = 0|\mathbf{z}^{1:n}, \mathbf{x}) \prod_{i=1}^{n} q_\phi(\mathbf{z}^i, z_{\mathrm{pres}}^i = 1|\mathbf{x}, \mathbf{z}^{1:i-1}). \tag{2}$$

$q_\phi$ is implemented as a neural network that, in each step, outputs the parameters of the sampling distributions over the latent variables, e.g., the mean and standard deviation of a Gaussian distribution for continuous variables. $z_{\mathrm{pres}}$ can be understood as an interruption variable: at each time step, if the network outputs $z_{\mathrm{pres}} = 1$, it describes at least one more object and proceeds, but if it outputs $z_{\mathrm{pres}} = 0$, no more objects are described, and inference terminates for that particular datapoint.

Note that conditioning of $\mathbf{z}^i|\mathbf{x}, \mathbf{z}^{1:i-1}$ is critical to capture dependencies between the latent variables $\mathbf{z}^i$ in the posterior, e.g., to avoid explaining the same object twice. The specifics of the networks that achieve this depend on the particularities of the models and we will describe them in detail in Sec. 3.

## 2.2 Learning

We can jointly optimize the parameters $\theta$ of the model and $\phi$ of the inference network by maximizing the lower bound on the marginal likelihood of an image under the model: $\log p_\theta(\mathbf{x}) \geq \mathcal{L}(\theta, \phi) = \mathbb{E}_{q_\phi}\left[\log \frac{p_\theta(\mathbf{x}, \mathbf{z}, n)}{q_\phi(\mathbf{z}, n, |\mathbf{x})}\right]$ with respect $\theta$ and $\phi$. $\mathcal{L}$ is called the negative free energy. We provide an outline of how to construct an estimator of the gradient of this quantity below, for more details see [23].

Computing a Monte Carlo estimate of $\frac{\partial}{\partial\theta}\mathcal{L}$ is relatively straightforward: given a sample from the approximate posterior $(\mathbf{z}, \mathbf{z}_{\mathrm{pres}}) \sim q_\phi(\cdot|\mathbf{x})$ (i.e., when the latent variables have been 'filled in') we can readily compute $\frac{\partial}{\partial\theta} \log p_\theta(\mathbf{x}, \mathbf{z}, n)$ provided $p$ is differentiable in $\theta$.

Computing a Monte Carlo estimate of $\frac{\partial}{\partial \phi}\mathcal{L}$ is more involved. As discussed above, the RNN that implements $q_\phi$ produces the parameters of the sampling distributions for the scene variables $\mathbf{z}$ and presence variables $\mathbf{z}_{\text{pres}}$. For a time step $i$, denote with $\omega^i$ all the parameters of the sampling distributions of variables in $(z_{\text{pres}}^i, \mathbf{z}^i)$. We parameterize the dependence of this distribution on $\mathbf{z}^{1:i-1}$ and $\mathbf{x}$ using a recurrent function $R_\phi(\cdot)$ implemented as a neural network such that $(\omega^i, \mathbf{h}^i) = R_\phi(\mathbf{x}, \mathbf{h}^{i-1})$ with hidden variables $\mathbf{h}$. The full gradient is obtained via chain rule: $\partial\mathcal{L}/\partial\phi = \sum_i \partial\mathcal{L}/\partial\omega^i \times \partial\omega^i/\phi$. Below we explain how to compute $\partial\mathcal{L}/\partial\omega^i$. We first rewrite our cost function as follows: $\mathcal{L}(\theta, \phi) = \mathbb{E}_{q_\phi}[\ell(\theta, \phi, \mathbf{z}, n)]$ where $\ell(\theta, \phi, \mathbf{z}, n)$ is defined as $\log \frac{p_\theta(\mathbf{x}, \mathbf{z}, n)}{q_\phi(\mathbf{z}, n, |\mathbf{x})}$. Let $z^i$ be an arbitrary element of the vector $(\mathbf{z}^i, z_{\text{pres}}^i)$ of type {what, where, pres}. How to proceed depends on whether $z^i$ is continuous or discrete.

**Continuous:** Suppose $z^i$ is a continuous variable. We use the path-wise estimator (also known as the 're-parameterization trick', e.g., [9, 23]), which allows us to 'back-propagate' through the random variable $z^i$. For many continuous variables (in fact, without loss of generality), $z^i$ can be sampled as $h(\xi, \omega^i)$, where $h$ is a deterministic transformation function, and $\xi$ a random variable from a fixed noise distribution $p(\xi)$ giving the gradient estimate: $\frac{\partial\mathcal{L}}{\partial\omega^i} \approx \partial\ell(\theta, \phi, \mathbf{z}, n)/\partial z^i \times \partial h/\partial\omega^i$.

**Discrete:** For discrete scene variables (e.g., $z_{\text{pres}}^i$) we cannot compute the gradient $\partial\mathcal{L}/\partial\omega_j^i$ by back-propagation. Instead we use the likelihood ratio estimator [18, 23]. Given a posterior sample $(\mathbf{z}, n) \sim q_\phi(\cdot|\mathbf{x})$ we can obtain a Monte Carlo estimate of the gradient: $\partial\mathcal{L}/\partial\omega^i \approx \partial \log q(z^i|\omega^i)/\partial\omega^i \, \ell(\theta, \phi, \mathbf{z}, n)$. In the raw form presented here this gradient estimate is likely to have high variance. We reduce its variance using appropriately structured neural baselines [18] that are functions of the image and the latent variables produced so far.

## 3 Models and Experiments

We first apply AIR to a dataset of multiple MNIST digits, and show that it can reliably learn to detect and generate the constituent digits from scratch (Sec. 3.1). We show that this provides advantages over state-of-the-art generative models such as DRAW [3] in terms of computational effort, generalization to unseen datasets, and the usefulness of the inferred representations for downstream tasks. We also apply AIR to a setting where a 3D renderer is specified in advance. We show that AIR learns to use the renderer to infer the counts, identities and poses of multiple objects in synthetic and real table-top scenes with unprecedented speed (Sec. 3.2 and appendix J).

Details of the AIR model and networks used in the 2D experiments are shown in Fig. 2. The generative model (Fig. 2, left) draws $n \sim \text{Geom}(\rho)$ digits $\{\mathbf{y}_{\text{att}}^i\}$, scales and shifts them according to $\mathbf{z}_{\text{where}}^i \sim \mathcal{N}(0, \Sigma)$ using spatial transformers, and sums the results $\{y^i\}$ to form the image. Each digit is obtained by first sampling a latent code $\mathbf{z}_{\text{what}}^i$ from the prior $\mathbf{z}_{\text{what}}^i \sim \mathcal{N}(\mathbf{0}, \mathbf{1})$ and propagating it through a decoder network. The learnable parameters of the generative model are the parameters of this decoder network. The AIR inference network (Fig. 2, middle) produces three sets of variables for each entity at every time-step: a 1-dimensional Bernoulli variable indicating the entity's presence, a $C$-dimensional distributed vector describing its class or appearance ($\mathbf{z}_{\text{what}}^i$), and a 3-dimensional vector specifying the affine parameters of its position and scale ($\mathbf{z}_{\text{where}}^i$). Fig. 2 (right) shows the interaction between the inference and generation networks at every time-step. The inferred pose is used to attend to a part of the image (using a spatial transformer) to produce $\mathbf{x}_{\text{att}}^i$, which is processed to produce the inferred code $\mathbf{z}_{\text{code}}^i$ and the reconstruction of the contents of the attention window $\mathbf{y}_{\text{att}}^i$. The same pose information is used by the generative model to transform $\mathbf{y}_{\text{att}}^i$ to obtain $\mathbf{y}^i$. This contribution is only added to the canvas $\mathbf{y}$ if $z_{\text{pres}}^i$ was inferred to be true.

For the dataset of MNIST digits, we also investigate the behavior of a variant, difference-AIR (DAIR), which employs a slightly different recurrent architecture for the inference network (see Fig. 8 in appendix). As opposed to AIR which computes $\mathbf{z}^i$ via $\mathbf{h}^i$ and $\mathbf{x}$, DAIR reconstructs at every time step $i$ a partial reconstruction $\mathbf{x}^i$ of the data $\mathbf{x}$, which is set as the mean of the distribution $p_\theta^x(\mathbf{x}|\mathbf{z}^1, \mathbf{z}^2, \ldots, \mathbf{z}^{i-1})$. We create an error canvas $\Delta\mathbf{x}^i = \mathbf{x}^i - \mathbf{x}$, and the DAIR inference equation $R_\phi$ is then specified as $(\omega^i, \mathbf{h}^i) = R_\phi(\Delta\mathbf{x}^i, \mathbf{h}^{i-1})$.

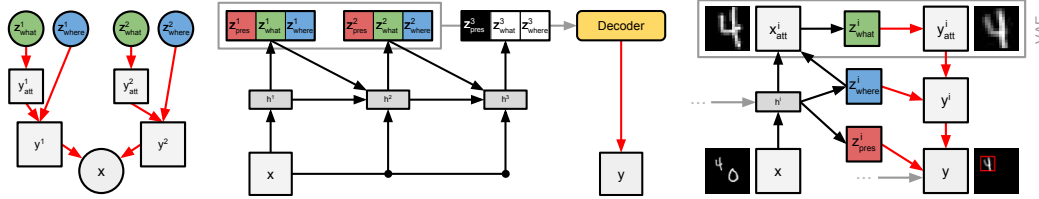

Figure 2: **AIR in practice:** *Left:* The assumed generative model. *Middle:* AIR inference for this model. The contents of the grey box are input to the decoder. *Right:* Interaction between the inference and generation networks at every time-step. In our experiments the relationship between $\mathbf{x}_{att}^i$ and $\mathbf{y}_{att}^i$ is modeled by a VAE, however any generative model of patches could be used (even, e.g., DRAW).

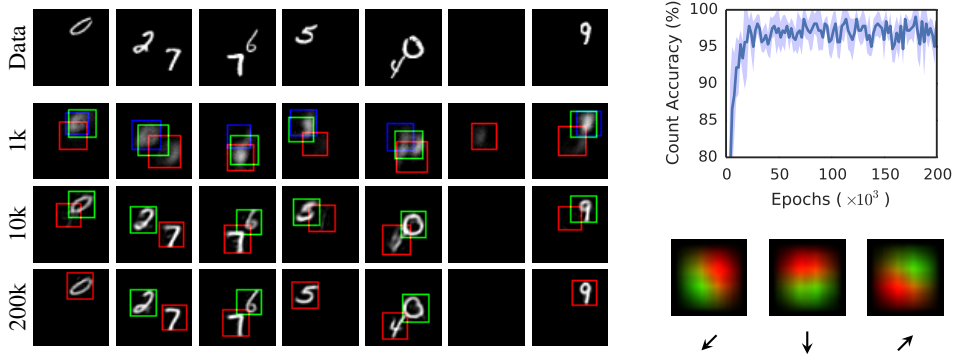

Figure 3: **Multi-MNIST learning:** *Left above:* Images from the dataset. *Left below:* Reconstructions at different stages of training along with a visualization of the model's attention windows. The 1st, 2nd and 3rd time-steps are displayed using red, green and blue borders respectively. A video of this sequence is provided in the supplementary material. *Above right:* Count accuracy over time. The model detects the counts of digits accurately, despite having never been provided supervision. Chance accuracy is 25%. *Below right:* The learned scanning policy for 3 different runs of training (only differing in the random seed). We visualize empirical heatmaps of the attention windows' positions (red, and green for the first and second time-steps respectively). As expected, the policy is random. This suggests that the policy is spatial, as opposed to identity- or size-based.

## 3.1 Multi-MNIST

We begin with a $50 \times 50$ dataset of multi-MNIST digits. Each image contains zero, one or two non-overlapping random MNIST digits with equal probability. The desired goal is to train a network that produces sensible explanations for each of the images. We train AIR with $N = 3$ on 60,000 such images from scratch, i.e., without a curriculum or any form of supervision by maximizing $\mathcal{L}$ with respect to the parameters of the inference network and the generative model. Upon completion of training we inspect the model's inferences (see Fig. 3, left). We draw the reader's attention to the following observations. First, the model identifies the number of digits correctly, due to the opposing pressures of (a) wanting to explain the scene, and (b) the cost that arises from instantiating an object under the prior. This is indicated by the number of attention windows in each image; we also plot the accuracy of count inference over the course of training (Fig. 3, above right). Second, it locates the digits accurately. Third, the recurrent network learns a suitable scanning policy to ensure that different time-steps account for different digits (Fig. 3, below right). Note that we did not have to specify any such policy in advance, nor did we have to build in a constraint to prevent two time-steps from explaining the same part of the image. Finally, that the network learns to not use the second time-step when the image contains only a single digit, and to never use the third time-step (images contain a maximum of two digits). This allows for the inference network to stop upon encountering the first $z_{pres}^i$ equaling 0, leading to potential savings in computation during inference.

A video showing real-time inference using AIR has been included in the supplementary material. We also perform experiments on Omniglot ([13], appendix G) to demonstrate AIR's ability to parse glyphs into elements resembling 'strokes', as well as a dataset of sprites where the scene's elements appear under significant overlap (appendix H). See appendices for details and results.

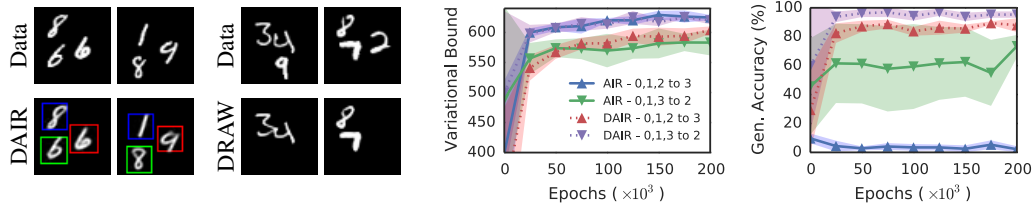

Figure 4: **Strong generalization:** *Left:* Reconstructions of images with 3 digits made by DAIR trained on 0, 1 or 2 digits, as well as a comparison with DRAW. *Right:* Variational lower bound, and generalizing / interpolating count accuracy. DAIR out-performs both DRAW and AIR at this task.

### 3.1.1 Strong Generalization

Since the model learns the concept of a digit independently of the positions or numbers of times it appears in each image, one would hope that it would be able to generalize, e.g., by demonstrating an understanding of scenes that have structural differences to training scenes. We probe this behavior with the following scenarios: (a) *Extrapolation:* training on images each containing 0, 1 or 2 digits and then testing on images containing 3 digits, and (b) *Interpolation:* training on images containing 0, 1 or 3 digits and testing on images containing 2 digits. The result of this experiment is shown in Fig. 4. An AIR model trained on up to 2 digits is effectively unable to infer the correct count when presented with an image of 3 digits. We believe this to be caused by the LSTM which learns during training never to expect more than 2 digits. AIR's generalization performance is improved somewhat when considering the interpolation task. DAIR by contrast generalizes well in both tasks (and finds interpolation to be slightly easier than extrapolation). A closely related baseline is the Deep Recurrent Attentive Writer (DRAW, [3]), which like AIR, generates data sequentially. However, DRAW has a fixed and large number of steps (40 in our experiments). As a consequence generative steps do not correspond to easily interpretable entities, complex scenes are drawn faster and simpler ones slower. We show DRAW's reconstructions in Fig. 4. Interestingly, DRAW learns to ignore precisely one digit in the image. See appendix for further details of these experiments.

### 3.1.2 Representational Power

A second motivation for the use of structured models is that their inferences about a scene provides useful representations for downstream tasks. We examine this ability by first training an AIR model on 0, 1 or 2 digits and then produce inferences for a separate collection of images that contains precisely 2 digits. We split this data into training and test and consider two tasks: (a) predicting the sum of the two digits (as was done in [1]), and (b) determining if the digits appear in an ascending order. We compare with a CNN trained from the raw pixels, as well as interpretations produced by a convolutional

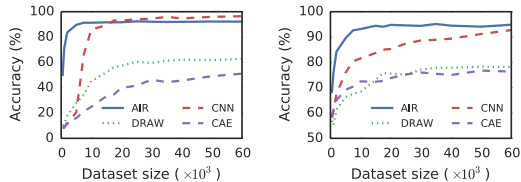

Figure 5: **Representational power:** AIR achieves high accuracy using only a fraction of the labeled data. *Left:* summing two digits. *Right:* detecting if they appear in increasing order. Despite producing comparable reconstructions, CAE and DRAW inferences are less interpretable than AIR's and therefore lead to poorer downstream performance.

autoencoder (CAE) and DRAW (Fig. 5). We optimize each model's hyper-parameters (e.g. depth and size) for maximal performance. AIR achieves high accuracy even when data is scarce, indicating the power of its disentangled, structured representation. See appendix for further details.

## 3.2 3D Scenes

The experiments above demonstrate learning of inference *and* generative networks in models where we impose structure in the form of a variable-sized representation and spatial attention mechanisms. We now consider an additional way of imparting knowledge to the system: we specify the generative model via a 3D renderer, i.e., we completely specify how any scene representation is transformed to produce the pixels in an image. Therefore the task is to learn to infer the counts, identities and poses of several objects, given different images containing these objects and an implementation of a 3D renderer from which we can draw new samples. This formulation of computer vision is often called 'vision as inverse graphics' (see e.g., [4, 15, 7]).

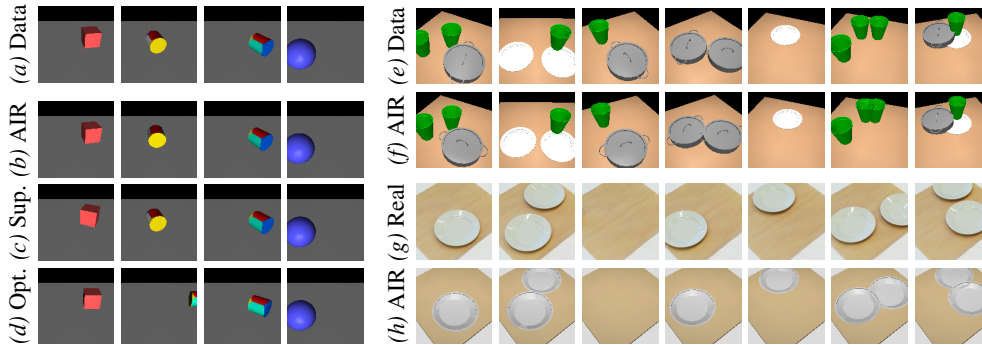

Figure 6: **3D objects:** *Left:* The task is to infer the identity and pose of a single 3D object. (a) Images from the dataset. (b) Unsupervised AIR reconstructions. (c) Supervised reconstructions. Note poor performance on cubes due to their symmetry. (d) Reconstructions after direct gradient descent. This approach is less stable and much more susceptible to local minima. *Right:* AIR can learn to recover the counts, identities and poses of multiple objects in a 3D table-top scene. (e,g) Generated and real images. (f,h) AIR produces fast and accurate inferences which we visualize using the renderer.

The primary challenge in this view of computer vision is that of inference. While it is relatively easy to specify high-quality models in the form of probabilistic renderers, posterior inference is either extremely expensive or prone to getting stuck in local minima (e.g., via optimization or MCMC). In addition, probabilistic renderers (and in particular renderers) typically are not capable of providing gradients with respect to their inputs, and 3D scene representations often involve discrete variables, e.g., mesh identities. We address these challenges by using finite-differencing to obtain a gradient through the renderer, using the score function estimator to get gradients with respect to discrete variables, and using AIR inference to handle correlated posteriors and variable-length representations.

We demonstrate the capabilities of this approach by first considering scenes consisting of only one of three objects: a red cube, a blue sphere, and a textured cylinder (see Fig. 6a). Since the scenes only consist of single objects, the task is only to infer the identity (cube, sphere, cylinder) and pose (position and rotation) of the object present in the image. We train a single-step ($N = 1$) AIR inference network for this task. The network is only provided with unlabeled images and is trained to maximize the likelihood of those images under the model specified by the renderer. The quality of the inferred scene representations produced is visually inspected in Fig. 6b. The network accurately and reliably infers the identity and pose of the object present in the scene. In contrast, an identical network trained to predict the ground-truth identity and pose values of the training data (in a similar style to [11]) has much more difficulty in accurately determining the cube's orientation (Fig. 6c). The supervised loss forces the network to predict the exact angle of rotation. However this is not identifiable from the image due to rotational symmetry, which leads to conditional probabilities that are multi-modal and difficult to represent using standard network architectures. We also compare with direct optimization of the likelihood from scratch for every test image (Fig. 6d), and observe that this method is slower, less stable and more susceptible to local minima. So not only does amortization reduce the cost of inference, but it also overcomes the pitfalls of independent gradient optimization.

We finally consider a more complex setup, where we infer the counts, identities and positions of a variable number of crockery items, as well as the camera position, in a table-top scene. This would be of critical importance to a robot, say, which is tasked with clearing the table. The goal is to learn to perform this task with as little supervision as possible, and indeed we observe that with AIR it is possible to do so with no supervision other than a specification of the renderer. We show reconstructions of AIR's inferences on generated data, as well as real images of a table with varying numbers of plates, in Fig. 6 and Fig. 7. AIR's inferences of counts, identities and positions are accurate for the most part. For transfer to real scenes we perform random color and size pertubations to rendered objects during training, however we note that robust transfer remains a challenging problem in general. We provide a quantitative comparison of AIR's inference robustness and accuracy on generated scenes with that of a fully supervised network in Fig. 7. We consider two scenarios: one where each object type only appears exactly once, and one where objects can repeat in the scene. A naive supervised setup struggles with object repetitions or when an arbitrary ordering of the objects is imposed by the labels, however training is more straightforward when there are no repetitions. AIR achieves competitive reconstruction and counts despite the added difficulty of object repetitions.

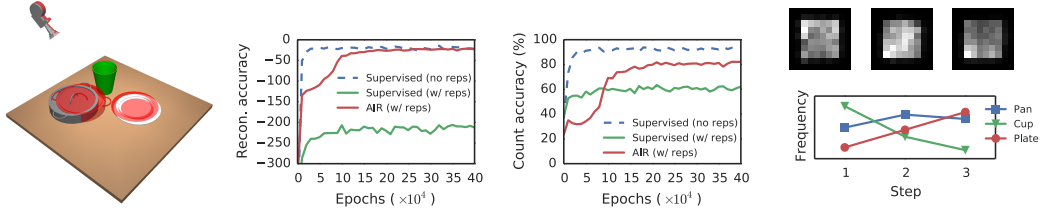

Figure 7: **3D scenes details:** *Left:* Ground-truth object and camera positions with inferred positions overlaid in red (note that inferred cup is closely aligned with ground-truth, thus not clearly visible). We demonstrate fast inference of all relevant scene elements using the AIR framework. *Middle:* AIR produces significantly better reconstructions and count accuracies than a supervised method on data that contains repetitions, and is even competitive on simpler data. *Right:* Heatmap of object locations at each time-step (top). The learned policy appears to be more dependent on identity (bottom).

## 4 Related Work

Deep neural networks have had great success in learning to predict various quantities from images, e.g., object classes [10], camera positions [8] and actions [20]. These methods work best when large labeled datasets are available for training. At the other end of the spectrum, e.g., in 'vision as inverse graphics', only a generative model is specified in advance and prediction is treated as an inference problem, which is then solved using MCMC or message passing at test-time. These models range from highly specified [17, 16], to partially specified [28, 24, 25], to largely unspecified [22]. Inference is very challenging and almost always the bottle-neck in model design.

Several works exploit data-driven predictions to empower the 'vision as inverse graphics' paradigm [5, 7]. For instance, in PICTURE [11], the authors use a deep network to distill the results of slow MCMC, speeding up predictions at test-time. Variational auto-encoders [21, 9] and their discrete counterparts [18] made the important contribution of showing how the gradient computations for learning of amortized inference and generative models could be interleaved, allowing both to be learned simultaneously in an end-to-end fashion (see also [23]). Works like that of [12] aim to learn disentangled representations in an auto-encoding framework using special network structures and / or careful training schemes. It is also worth noting that attention mechanisms in neural networks have been studied in discriminative and generative settings, e.g., [19, 6, 3].

AIR draws upon, extends and links these ideas. By its nature AIR is also related to the following problems: counting [14, 27], pondering [2], and gradient estimation through renderers [15]. It is the combination of these elements that unlocks the full capabilities of the proposed approach.

## 5 Discussion

In this paper our aim has been to learn unsupervised models that are good at scene understanding, in addition to scene reconstruction. We presented several principled models that learn to count, locate, classify and reconstruct the elements of a scene, and do so in a fraction of a second at test-time. The main ingredients are (a) building in meaning using appropriate structure, (b) amortized inference that is attentive, iterative and variable-length, and (c) end-to-end learning.

We demonstrated that model structure can provide an important inductive bias that gives rise to interpretable representations that are not easily learned otherwise. We also showed that even for sophisticated models or renderers, fast inference is possible. We do not claim to have found an ideal model for all images; many challenges remain, e.g., the difficulty of working with the reconstruction loss and that of designing models rich enough to capture all natural factors of variability.

Learning in AIR is most successful when the variance of the gradients is low and the likelihood is well suited to the data. It will be of interest to examine the scaling of variance with the number of objects and alternative likelihoods. It is straightforward to extend the framework to semi- or fully-supervised settings. Furthermore, the framework admits a plug-and-play approach where existing state-of-the-art detectors, classifiers and renderers are used as sub-components of an AIR inference network. We plan to investigate these lines of research in future work.

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
