[Supplementary Material · supplementary.pdf]

# A  Stochastic Gradient Estimators

In this section, we give further details behind the equations in Sec. 2. We simplify notation by not referencing the model parameters $\theta$ and considering a single latent $z$ at a time. Assume we have a function $\ell(z)$ and distribution $q_\phi(z)$; we wish to estimate $\nabla_\phi \mathbb{E}[\ell(z)]$.

## A.1  Reparameterization trick

As per the main body, we supposed the existence of a differentiable function $h$ and random variable $\xi$ with fixed noise distribution $p_\xi(\cdot)$ such that $h(\xi, \phi) \sim q_\phi(\cdot)$. It follows that:

$$
\begin{aligned}
\frac{\partial}{\partial \phi} \mathbb{E}_{z \sim q_\phi}[\ell(z)] &= \frac{\partial}{\partial \phi} \mathbb{E}_{\xi \sim p_\xi}[\ell(h(\xi, \phi))] \\
&= \mathbb{E}_{\xi \sim p_\xi}\left[\frac{\partial}{\partial \phi} \ell(h(\xi, \phi))\right] \\
&= \mathbb{E}_{\xi \sim p_\xi}\left[\frac{\partial \ell}{\partial z} \frac{\partial h}{\partial \phi}\right] \\
&= \mathbb{E}_{z \sim q_\phi}\left[\frac{\partial \ell}{\partial z} \frac{\partial h}{\partial \phi}\right] \\
&\approx \frac{\partial \ell(z)}{\partial z} \frac{\partial h(\xi, \phi)}{\partial \phi}.
\end{aligned}
\tag{3}
$$

In other words, an estimate of the gradient can be recovered by forwarding sampling the model by using the reparameterization given by h, and backpropagating normally through $h$.

## A.2  Likelihood ratio estimator

The likelihood ratio method simply uses the equality:

$$
\frac{\partial \log q_\phi(z)}{\partial \phi} = \frac{\dfrac{\partial q_\phi(z)}{\partial \phi}}{q_\phi(z)}
\tag{4}
$$

to rewrite an integral as an expectation. Assuming that $\frac{\partial q_\phi(z)}{\partial \phi}$ exists and is continuous, we have:

$$
\begin{aligned}
\frac{\partial}{\partial \phi} \int q_\phi(z)\ell(z)\partial z &= \int_z \frac{\partial q_\phi(z)}{\partial \phi} q_\phi(z)\mathrm{d}z \\
&= \int_z \frac{\partial \log \ell_\phi(z)}{\partial \theta} \ell_\phi(z)\ell(z)\mathrm{d}z \\
&= \mathbb{E}_{q_\phi(z)}\left[\frac{\partial \log q_\phi(z)}{\partial \phi} \ell(z)\right] \\
&\approx \frac{\partial \log q_\phi(z)}{\partial \phi} \ell(z).
\end{aligned}
\tag{5}
$$

Note that if $\ell(z)$ is a constant with respect to $z$, then the expression is clearly 0, since the integral evaluates to the same constant.

# B  Prior for Unary Encoding

Recall that we can encode the number of objects $n$ as a variable length unary code vector $\mathbf{z}_{\text{pres}}$ defined by $z_{\text{pres}}^i = 1$ for $i \leq n$, and $z_{\text{pres}}^{n+1} = 0$ (more generally, it can be useful to implicitly define $z_{\text{pres}}^j = 0$, for $j > n$). Consider an arbitrary distribution $p(\cdot)$ over $n$, and denote $\mu_{\geq n} = \sum_{k \geq n} p(k)$ the probability that there are at least $n$ objects. We define a joint probability distribution for $\mathbf{z}_{\text{pres}}$ and show it is consistent with $p(n)$.

Figure 8: **AIR vs. DAIR:** *Left:* The standard AIR architecture. *Right:* The DAIR architecture. At each time-step $i$, the latent variables produced so far are used to perform a partial rendering of the scene. The difference of this partial rendering from the image under question is used infer $z_{\text{pres}}^i$, $\mathbf{z}_{\text{what}}^i$ and $\mathbf{z}_{\text{where}}^i$ in the current time-step.

Let $p(z_{\text{pres}}^i = 1 | z_{\text{pres}}^{i-1}) = z_{\text{pres}}^{i-1} \frac{\mu_{\geq i}}{\mu_{\geq(i-1)}}$ for $i \geq 2$, and $p(z_{\text{pres}}^1) = \mu_{\geq 1}$. Note that if $z_{\text{pres}}^i = 0$ for any $i$, it follows immediately that $z_{\text{pres}}^j = 0$ for $j \geq i$. The sampled vector is therefore a correct unary code. Furthermore,

$$
\begin{aligned}
&P(\max\{i : z_{\text{pres}}^i = 1\} = n) \\
&= P(z_{\text{pres}}^1 = 1, z_{\text{pres}}^2 = 1, \ldots, z_{\text{pres}}^n = 1, z_{\text{pres}}^{n+1} = 0) \\
&= \left( \prod_{i=1}^{n} P(z_{\text{pres}}^i = 1 | z_{\text{pres}}^{i-1} = 1) \right) P(z_{\text{pres}}^{n+1} = 0 | z_{\text{pres}}^n = 1) \\
&= \mu_{\geq 1} \times \frac{\mu_{\geq 2}}{\mu_{\geq 1}} \times \frac{\mu_{\geq 3}}{\mu_{\geq 2}} \cdots \frac{\mu_{\geq n}}{\mu_{\geq(n-1)}} \times \left( 1 - \frac{\mu_{\geq(n+1)}}{\mu_{\geq n}} \right) \\
&= \mu_{\geq n} - \mu_{\geq(n+1)} \\
&= p(n)
\end{aligned}
$$

It follows that for $\mathbf{z}_{\text{pres}}$ following the distribution specified above, the corresponding maximum index is distributed according to $p(n)$ as desired.

## C   Details of 2D Experiments

All experiments were performed with a batch size of 64. Inference networks and decoders were trained using a learning rate of $10^{-4}$ and baselines were trained using a higher learning rate of $10^{-3}$. LSTMs had 256 cell units and object appearances were coded with 50 units. Images were normalized to hold values between 0 and 1 and the likelihood function was a Gaussian with fixed standard deviation equal to 0.3. The prior $p(n)$ was fixed to a geometric distribution which favors sparse reconstructions.

## D   Details of the DAIR Network

We assume that the renderer likelihood $p^x(\mathbf{x}|\mathbf{z}^1, \mathbf{z}^2, \ldots, \mathbf{z}^i)$ has a link function $I$ which maps a sufficient statistic $h^i$ to the mean; $h^i$ can be iteratively updated from $h^{i-1}$ and $z^{i-1}$. this is the case for instance for Gaussian and Bernoulli distributions (where $h^i$ is respectively taken to be the mean and log-odds of the distribution). In DAIR, we use the error $\Delta x^i$ between the partial reconstruction $I(h^{i-1})$ and the data $\mathbf{x}$ as inputs to a feed-forward neural network which predicts $\mathbf{z}^i, z_{\text{pres}}^i$. DAIR can be thought of as a special case of AIR with additional structure; namely, the recurrent aspect of AIR is fixed to become a canvas-reconstruction network; see Fig. 8 for more details.

## E   Details of AIR vs. CNN vs. CAE vs. DRAW Experiments

The convolutional neural network uses a $64\times(5\times5)$-$64\times(5\times5)$-$64\times(5\times5)$-$512$ architecture.

| Model | Free Energy | |
|---|---|---|
| | Up to 2 digits | Only 3 digits |
| DRAW | $-\mathbf{637}$ | $-406$ |
| AIR | $-620$ | $-316$ |
| DAIR | $-611$ | $-\mathbf{424}$ |

Table 1: **Comparisons with state-of-the-art.** DRAW achieves lower scores than AIR and DAIR on up to 2 digits but is outperformed by DAIR when generalizing to 3 digits.

Figure 9: **Omniglot:** AIR reconstructions at every time-step. AIR uses variable numbers of strokes for digits of varying complexity.

The convolutional autoencoder uses a sequence of 3 $64 \times (6 \times 6)$ (for slightly increased performance over $5 \times 5$ filters) convolutions with $2 \times 2$ max-pooling layers for the encoding, and 3 full convolutions (of the same sizes) and a $2 \times 2$ nearest neighbor upsampler for the deconvolution.

The embeddings created by AIR, DRAW, or CAE are fed through a 4-layer network (each with $512$ units) to produce the 19-way prediction of the sum or a 2-way prediction of the order.

## F    DRAW Comparisons

We compare AIR and DAIR to a state of the art DRAW network with $40$ drawing steps with $4$ latent units per time step, $400$ LSTM hidden units, spatial transformer [6] attention module, and single read and write heads of size $16 \times 16$. We report free energy on two test sets: a test dataset with $0$, $1$ or $2$ digits, and another with images with precisely $3$ digits. The likelihood model was in all cases Gaussian with fixed standard deviation of $0.3$. DRAW outperforms AIR and DAIR on the $0/1/2$ dataset; this is likely due to the fact that DRAW uses many more drawing steps ($40$) than AIR and thus has an excellent statistical model of single digits. DRAW however does not conceptually understand them as distinct units, as evidenced by its poor generalization on the 3-digits dataset, where DAIR has both better score, and more meaningful reconstruction: DAIR partially generalizes to a number of digit never seen (Fig. 4), while DRAW interestingly learns to perfectly ignore exactly one digit in the image (see Fig. 4). More generally, the VAE subroutine present in AIR could be replaced by a DRAW network, thus leading to a 'best of both worlds' model with excellent single digit model and understanding of a scene in terms of its constituent parts.

## G    Omniglot Experiments

We also investigate the behavior of AIR on the Omniglot dataset [13] which contains 1623 different handwritten characters from 50 different alphabets. Each of the 1623 characters was drawn online via Amazon's Mechanical Turk by 20 people. This means that the data was produced according a process

Figure 10: **Sprites overview:** (a) Images from the dataset. (b) AIR reconstructions. We visualize the model's attention at every time-step (inferred object boundaries) in white. (c) A selection of samples from the learned model.

Figure 11: **Sprites quantitative results:** *Left:* Variational lower bound over the course of training. *Right:* Sprite count accuracy.

(pen strokes) that is not directly reflected in the structure of our generative model. It is therefore interesting to examine the outcome of learning under mis-specification. We train the model from the previous section, this time allowing for a maximum of up to 4 inference time-steps per image. Fig. 9 shows that by using different numbers of time-steps to describe characters of varying complexity, AIR discovers a representation consisting of spatially coherent elements resembling strokes, despite not exploiting stroke labels in the data or building in the physics of strokes, in contrast with [13]. Further results can be found in the supplementary video.

## H   Sprites Experiments

We also consider a $50 \times 50$ dataset of sprites: red circles, green squares and blue diamonds. Each image in the dataset contains zero, one or two sprites (see Fig. 10a). The images are composed additively (sprites do not occlude each other). We use the exact same model structure as for the multi-MNIST dataset.

At the end of unsupervised training, AIR successfully learns about the underlying causes of the scenes (namely, the sprites), as well as their counts and locations, and also produces convincing reconstructions (see Fig. 10b). Note that the inference network correctly detects the correct number of sprites even when two overlapping sprites of the same type and color appear in the same image (Fig. 10a,b, images 1 and 3). Also note that the reconstructions are accurate, meaning that the inference network successfully produces the codes for each sprite despite the presence of the other sprites in its field of view. Fig. 10c displays a collection of samples from the model after training. We display quantitative evaluation of the network's counting accuracy in Fig. 11, reconstructions over the course of learning in Fig. 12, and a visualization of its scanning policy in Fig. 13.

Note that these tasks can only be successfully achieved once the inference network has learned a sensible policy for scanning the image, e.g., one in which every object is attended to only once. However the network must break multiple symmetries to achieve this, e.g., it does not matter which object it explains first. In Fig. 13 we visualize the learned scanning policy for 3 different runs of training (only differing in the random seed). In each case a unique policy is learned, and the policy appears to be spatial (as opposed to one that is based on digit identity or size).

Figure 12: **Sprites learning:** *Top:* Images from the dataset. *Bottom:* Reconstructions at different points during training. A video of this sequence is included in the supplementary material.

Figure 13: **Sprites scanning policies:** Empirical heatmaps of where the attention windows go to (red, and green for the first and second time-steps respectively). As expected, the policy is random. Each figure is for a different inference network that has been trained from scratch using a different seed. This suggests that the policy is spatial, as opposed to identity- or size-based.

# I   Details of 3D Scene Experiments

The experiments in section 3.2 were performed using the rendering capabilities of the MuJoCo physics simulator [26].

## I.1   Gradient estimation

Differentiation of MuJoCo's graphics engine was performed using forward finite-differencing (with a constant $\epsilon = 10^{-4}$) with respect to the scene configuration. This is a generic procedure which would work for any graphics engine; we chose MuJoCo because it is fast (using only the fixed functionality of OpenGL) and because scenes are conveniently parameterized. Interestingly, despite the coarse 8-bit output of OpenGL, quantization errors appeared to average out reasonably well over the pixels.

## I.2   Scene generation

**Single object scenes:**   For the results shown in Fig. 6 we created a scene that contained a MuJoCo box geom representing the table, 3 'objects' (also in the form of MuJoCo geoms; cube, sphere, textured cylinder), and a fixed camera. The objects could be moved in the plane of the table and rotated along the axis orthogonal to it (i.e. 3 degrees of freedom per object). We created random scenes containing at most one object by randomly sampling position, rotation angle, object presence (visibility) and object type. (Geoms were made invisible by moving them out of the field of view of the camera.) An illustration is shown in Fig. 14.

**Tabletop scenes:**   For the results shown in Fig. 6 and 7 we used scenes with a box geom for the table, and nine mesh geoms for the crockery items. The cup, pan, and plate were each replicated three times to allow for arbitrary three-objects scenes. Each geom had three degrees of freedom (position in the table plane and rotation). Random scenes with up to $N = 3$ objects were created by uniformly

Figure 14: *Left*: Illustration of the setup for single-object scenes similar to Fig. 7 in the main text. The illustration shows the fixed camera, the ground truth object (textured cylinder), and an example inference (transparent red). *Right*: Rendering from camera as fed into the inference network (before downsampling).

sampling position, rotation angle, object presence, and object type three times. As for the single objects were rendered invisible by moving them outside of the field of view of the camera.

We experimented with two versions of the scene: one with a fixed camera, and one version where the camera could be moved in an orbit around the table (i.e. one degree of freedom). We discuss the experiment with the fixed camera in the main text. For the latter set of scenes, the camera position was also chosen randomly and the image was rendered from the random camera position. Camera movement was restricted to $\pm 40$ degrees from the central position. In this experiment the model had to learn to infer the camera position in addition to the objects on the table. The montage in Fig. 7 in the main text shows a ground truth scene (with camera) and the inferred identities and positions of the objects as well as the inferred position of the camera. We show several examples of random scenes with variable camera and the associated inferences in Fig. 15. For the most part the network infers all scene parameters reliably.

**Image preprocessing:** We rendered all scene images at $128 \times 128$ pixels. We down-sampled scene images to $32 \times 32$ pixels for input to the network.

### I.3  Model

We trained a network to perform inference in the following fixed generative model:

$$p(\mathbf{x}, z_{\text{pres}}^{1:N}, \mathbf{z}_{\text{where}}^{1:N}, \mathbf{z}_{\text{what}}^{1:N}) = \tag{6}$$

$$p(\mathbf{x}|z_{\text{pres}}^{1:N}, \mathbf{z}_{\text{where}}^{1:N}, \mathbf{z}_{\text{what}}^{1:N}) \prod_{i=1}^{N} p(z_{\text{pres}}^i) p(\mathbf{z}_{\text{what}}^i) p(\mathbf{z}_{\text{where}}^i),$$

where $z_{\text{pres}}^i$ is the visibility indicator: $z_{\text{pres}}^i \sim \text{Bernoulli}(\alpha)$ for object $i$; $\mathbf{z}_{\text{where}} \in \mathbb{R}^3$ indicates position and rotation angle: $\mathbf{z}_{\text{where}}^i \sim \mathcal{N}(0, \Sigma_{\text{where}})$; and $\mathbf{z}_{\text{what}}^i$ is a three-valued discrete variable indicating the object type (mesh / geom type): $\mathbf{z}_{\text{what}}^i \sim \text{Discrete}(\beta)$.

The marginal distribution over scenes under this model is the same as the marginal distribution under a model of form described in Section 2 in the main text where $p(n) = \text{Binomial}(N, \alpha)$ and $n = \sum_{i=1}^{N} z_i$.

For the variable camera scenes the model included an additional random variable $z_{\text{cam}} \in \mathbb{R}$ where $z_{\text{cam}} \sim \mathcal{N}(0, \sigma_{\text{cam}}^2)$.

To evaluate the likelihood term $p(\mathbf{x}|\mathbf{z})$ we (1) render the scene description using the MuJoCo rendering engine to produce a high-resolution image $\mathbf{y}$; (2) blur the resulting image $\mathbf{y}$ as well as $\mathbf{x}$ using a fixed-with blur kernel; (3) compute $\mathcal{N}(\mathbf{x}|\mathbf{y}, \mathbf{I}\sigma_x^2)$.

Figure 15: **3D scenes with variable camera:** AIR results for inferring the camera angle of the scene, as well as the counts, identities and poses of multiple objects in a 3D table-top scene similar to the results presented in Section 3.2 in the main text but with the additional complication of an unknown camera angle. (a) Images from the dataset. (b) Reconstruction of the scene description inferred by our AIR network. Note that due to the down-sampling of the images that were used as input to the inference network and the blurring in the likelihood computation accurate estimation of the rotation angle is essentially impossible.

## I.4   Network

The AIR inference network for our experiments is a standard recurrent network (no LSTM) that is run for a fixed number of steps ($N = 1$ or $N = 3$). In each step the network computes:

$$(\omega^i_{\text{pres}}, \omega^i_{\text{what}}, \omega^i_{\text{where}}, \mathbf{h}^i) = R(\mathbf{x}, z^{i-1}_{\text{pres}}, \mathbf{z}^{i-1}_{\text{what}}, \mathbf{z}^{i-1}_{\text{where}}, \mathbf{h}^{i-1}),$$

where the $\omega^i$ represent the parameters of the sampling distributions for the random variables: Bernoulli for $z_{\text{pres}}$; Discrete for $\mathbf{z}_{\text{what}}$; and Gaussian for $\mathbf{z}_{\text{where}}$. For the experiments with random camera angle we use a separate network that computes $\omega_{\text{cam}} = F(\mathbf{x})$ and we provide the sampled camera angle as additional input to $R$ at each time step.

## I.5   Supervised learning

For the baselines trained in a supervised manner we use the ground truth scene variables $z^{1:N}_{\text{pres}}, \mathbf{z}^{1:N}_{\text{where}}, \mathbf{z}^{1:N}_{\text{what}}$ that underly the training scene images as labels and train a network of the same form as the inference network to maximize the conditional log likelihood of the ground truth scene variables given the image.

## J    Inference Speed

For the MNIST experiments, upon completion of training each inference step takes 5.6 milliseconds on average to execute on an Nvidia Quadro K4000 GPU (a step corresponds to inference of state for a single object), in other words up to around 17 milliseconds per image for images of 3 digits. Therefore running at around 59 frames per second, inference is significantly faster than real-time.

For 3D scenes, the equivalent numbers are around 2.3 milliseconds per step and 8 milliseconds per image (due to absence of spatial transformers) on a CPU. Gradient-based optimization is slower, taking 5 milliseconds per gradient step per object, and tens or hundreds of steps per image, depending on the choice of optimizer.

Training for the MNIST model converges in around 2 days, and in around 3 days for the 3D scenes.