[Reviews · NeurIPS 2016]

Reviewer 1

Summary

This paper presents a framework for "vision as inverse graphics" that combines structured generative models, inference networks, and end-to-end learning. A key element of the framework is the use of a trans-dimensional generative model that allows for the generation of a variable number of objects in the scene. The inference network that enables learning in the model uses a recurrent network to parse the scene one object at a time. This mechanism simultaneously parses the current object in terms of what/where information and predicts whether there is another object in the scene that remains to be parsed. The evaluation considers a number of 2D and 3D instances of the proposed model including a 2D model that directly learns object appearances and a 3D model that integrates a full 3D renderer. The representations learned by the framework are used for a variety of down-stream tasks including counting objects, where the performance is shown to outperform DRAW networks.

Qualitative Assessment

Review Update: Tank you for the additional information on computation time as well as clarifying the issue with repetitions. Technical quality: The technical contribution of the paper is defined at the level of a framework with modular parts and so is quite high-level as a result. The main components are the generative model (Eq 1), the recurrent inference network, and the use of variational learning. To the extent that technical details are provided for these components, they are correct. The bulk of the paper focuses on the construction of experiments and the analysis of the results. In general, the data sets and tasks are well designed in both the 2D and 3D cases. The results in terms of object counts are promising. One aspect that could have been better evaluated in the 2D experiments is the accuracy of the pose (where) parameters inferred. This information is only presented qualitatively in terms of the reconstructions (although these do look good in the examples shown). In the case of 3D scenes, the results are also promising. However, the reconstruction accuracy metric appears not to be defined. The discussion of the problem caused by repetitions in the data sets is also not clear and should be explained further. A final important point regarding the experiments is that the paper claims to enable scene understanding at unprecedented speeds, but there are no timing results of any kind in the paper. The video in the supplemental material appears to be showing real-time scene parsing for multiple digits, but the speed results need to be directly substantiated in the main paper. Novelty/originality: As mentioned already, the paper describes a framework that combines several existing components: generative scene models, recurrent networks for inference, and variational learning. The type of generative model described has been studied in computer vision extensively in the vision as inverse graphics literature. This paper introduces a specific form of such a model that allows for inference of the number of objects and combines it with a recurrent inference network that can also reason about variable numbers of objects, which does appear to be a novel combination. The main novel claim made in the paper is enabling fast inference in this type of model, which has been the main issue in past work. However, as mentioned above, direct results substantiating the speed of the proposed framework are not included in the paper. Impact: As a framework, this work is also very extensible in different directions. The use of graphical models combined with RNN inference networks to enable parsing scenes with multiple objects is a nice idea and is likely to be adopted by others if the speed claims are suitably substantiated. Clarity and presentation: The paper is very well written. There are no issues with grammar or language. The paper is clear, although the descriptions are quite high-level for the most part. The figures are well done although most of the line graphs could benefit from being larger.

Confidence in this Review

2-Confident (read it all; understood it all reasonably well)


Reviewer 2

Summary

Summary: The paper proposes a method to perform efficient inference on structured image models that explicitly reason about objects. In the problem addressed by this work, structure from the shape (or form) in which the objects of interest occur independently. The proposed model discovers a specific number of scene elements and sequentially process (refines) them one a time via Recurent Neural Networks (RNNs). Experiments are coducted by considering 2D objects (Multi MNIST dataset) and 3D objects where it is shown that the proposed method is able to count, locate, and classify the objects of interest from images. Strong Points: - The criterion to stop the exploration of the image (Zpress==0 in Eq. (2)) seems to be novel. - The proposed method has a significant potential when introduced on works combining computer vision and computer graphics. Weak Points: - Experiments are performed on a relatively constrained setting. - The method seem to be sensitive to dataset bias when reasoning about object sizes/scales. (Failure|hard cases are rarely presented).

Qualitative Assessment

The proposed method achieves efficient inference for settings focusing on both the 2D image space or the 3D scene space. This last part promises a high potential of the proposed method on the increasing work that links the work from the computer vision and computer graphics communities. Some indications (and comparisons) of computation/processing times should motivate the adoption of the proposed method. Even thought the paper has no major flaw, in my opinion there are some aspects of the paper that should be improved: - From the conceptual point of view, the proposed method effectively achieves the goal of sequentially localizing a set of objects of interest. Furthermore, this is achieved by relying on very weak supervision. However, from the application point of view, the experiments presented on the paper were conducted on very constrained settings (eg. reduced number of objects, low level of clutter, reduced number of object classes). I encourage the authors to add experiments focusing on more realistic scenarios ( eg. see the very related work from Karlinsky et al. CVPR'10 or Gonzales-Garcia et al., CVPR'15 ). I guess some qualitative results on the performance of the proposed method on a more realistic setting might be enough. - From the Multi-MNIST Parts VS. whole visualizations presented in the supplementary material, it seems that the proposed method is sensitive to dataset bias ( eg. when the number 8 is too large it is considered as two different objects ). Based on this observation, I recommend adding and discussing some failure cases of the proposed methods in your evaluation. This should give a better picture on the cases where your method is expected to have more reliable performance and hint the situations under which it predictions are not so reliable. - In the evaluation, experiments are performed by assuming the maximum number of objects (N) to be lower or equal to three. I encourage the authors to add results when considering a higher number of objects. Moreover, in my opinion, the Extrapolation/Interpolation scenarios presented in Section 3.1.1. are of high relevance since they can give an indication of how sensitive is the performance of the method to the maximum number of objects (N) assumed during training. Unfortunately, this has received relatively low attention in the paper. I recommend reducing Section 3.2 and extend the analysis given in Section 3.1.1. - There is some missing information related to the experiments being performed. For completeness, it is important to provide information regarding the proportion of examples with a specific number of objects in the data, and an indication of what would be the accuracy levels obtained at random when reporting count accuracy (Fig 3,4,5,7). It is also important to describe how the multi-object examples were generated (what is at random?). In my opinion the description of the datasets can be further improved.

Confidence in this Review

2-Confident (read it all; understood it all reasonably well)


Reviewer 3

Summary

The paper introduces a recurrent neural network to identify and localize objects in an image by training the neural network through unsupervised learning, based on reconstructing the input image. The results in simple datasets show that the neural network effectivelly learns to identify and localize the objects in the images.

Qualitative Assessment

This paper tackles an important and challenging problem, which is reducing the level of supervision / sample complexity of neural networks for object recognition. The learning of the presented NN is based on reconstructing the input image, as in an autoencoder. About this, I think there are several assumptions that would be useful to clarify: -The structure of the NN introduced in the paper seems to be designed to accomodate the localization, counting, etc. of the different objects in the image. This structure is set at hand. It remains unclear what assumptions in the NN structure are necessary and which are not necessary to learn the tasks solved by the presented NN, and how do they help to reduce the sample complexity. -It would be very helpful some comments about any theoretical guarantees of the generalization properties of the presented method. In the results there is some emphasis about the generalization abilities of the presented method. Yet, it is unclear if some of these generalization abilities have been found from designing the NN by trial an error, or they come from some theoretical guidelines. I think it would help to be more convincing to put more emphasis on motivating the design of the NN from the generalisation properties, and in the experiments, an abaltion study would help better motivate the different parts of the NN. Finally, I give a low score for clarity of presentation because in several parts of the paper (eg. abstract and conclusions), some claims about the speed of the NN ("unprecedented speed") were not explained in the paper. Also, more complete explanations about the learning algorithm would make the reading of the paper more smooth, eg. why Eq.(1) is intractable, and why the adopted learning algorithm solves this intractability. In the discussion section it is mentioned that the model is end-to-end learning, but in the experiment for 3D scenes the renderer is not learned.

Confidence in this Review

2-Confident (read it all; understood it all reasonably well)


Reviewer 4

Summary

This paper presents a framework for scene interpretation with no supervision. The system detects multiple objects, identifies them and locates them. The framework is based on a generative model, and the inference is performed iteratively, using a recurrent neural network.

Qualitative Assessment

The paper presents and interesting approach for scene understanding from the perspective of 'vision as inverse graphics'. It is compared to DRAW, a system considered state-of-the-art under this paradigm. The results provided by the authors showed an improvement over DRAW in terms of generalization and computational cost. The structure of the paper is non-conventional. I would expect the related work section right after the introduction and a clearer description of the differences between previous methods and the presented approach. At the bottom of page 5 the authors refer to appendices for more details and results, but there are no appendices. I think the idea has a lot of potential in realistic scenarios (real images). Unfortunately the authors do not present any experiment on real images, integrating state-of-the-art detectors, classifiers, and renderers, as suggested in the Discussion. I think results on a real dataset would strength the paper.

Confidence in this Review

2-Confident (read it all; understood it all reasonably well)


Reviewer 5

Summary

The authors present a latent variable model and inference procedure for natural scene understanding. For a given image, they propose a variable number of latent variables, each of which (aim to) describe an object (or stroke/reusable feature). In order to jointly infer the latent variables of a scene jointly, the authors propose an iterative solver based on a recurrent neural network that describes the conditional distributions of the latent variables, given the image and past latent variables inferred. The latent variables themselves may encode class, location, and appearance information. Using an RNN as a recognition model, the authors apply their framework to a two-dimensional and three-dimensional (idealized) images. They show the ability of their model-inference procedure to identify the correct number of objects in a scene, and reconstruct object-specific attributes quickly.

Qualitative Assessment

This paper was a pleasure to read! Technical Quality * The method presented is technically sound, and the experiments and analysis are thorough and thoughtful. Novelty * This is the first paper I've seen that uses amortized variational inference to approach this type of structured inference. Potential Impact: * Like VAEs, this paper pushes the idea of using flexible/deep functions to approximate complicated conditional distributions. They augment this idea by using an iterative model for latent variables to model the posterior dependencies, which might be very impactful for future work on total scene understanding. Clarity: * The paper is very well written, ideas are presented clearly, and notation was carefully selected. Questions: - How does the iterative solver address the exchangeable symmetry of the problem that you mention in line 97? Is there a reason to believe that the inference procedure is invariant to the order in which objects are considered/properties inferred? - Figure 4 shows a dramatic difference in generalization accurate from the AIR => DAIR inference architecture. What is the reason or suspected reason for this? Is the generative model exactly the same, and the inference procedure is the only difference in this case?

Confidence in this Review

2-Confident (read it all; understood it all reasonably well)


Reviewer 6

Summary

The paper proposes a framework that extends the recently successful applications of (amortized) variational inference (e.g., variational auto-encoders). The framework allows to build upon previous approaches by imposing a probabilistic sequential inference mechanism: A variable length latent state is inferred step-wise via an RNN (i.e., object detection), and at each step, a "normal" VAE-like inference procedure is performed (i.e., object recognition). The network learns from data how many steps should be used.

Qualitative Assessment

The paper is a very good read: The motivation is immediately apparent. The architecture becomes clear at first read. The evaluation of the proposed architecture is abundant. The appeal of the proposed methodology is less based on deep theoretical insights, but on clever, well-designed combination of existing approaches (as is mentioned by the authors in section 4), including structured and unstructured components, that can easily be altered in each individual component (what the authors call "plug-and-play" approach in their conclusion). While technically all information is available with Suppl. Material, I would recommend shifting information that is crucially necessary for reproducibility (in particular appendices A and B) into the paper. Given that the analysis of the 3D experiments takes up a significant part of the paper, it is surprising to me that the difficulties with correctly rotating the pot are not analyzed. Apart from these minor criticisms, the paper will presumably influence many other researchers dealing with efficient inference. ------- Minor, easily fixable issues: - line 35: downstreams - lines 116-118: broken reference due to inline equation. - lines 126-128: hard-to-read equations due to line breaks.

Confidence in this Review

2-Confident (read it all; understood it all reasonably well)